# OPTIMIZING DYNAMIC TREATMENT STRATEGIES WITH REINFORCEMENT LEARNING AND DUAL-HAWKES PROCESS IN CLINICAL ENVIRONMENTS

## ABSTRACT

Modeling the timing of critical events and controlling associated risks through treatment options are crucial aspects of healthcare. However, current methods fall short in optimizing dynamic treatment plans to improve clinical outcomes. A key challenge lies in modeling the intensity functions of critical events throughout disease progression and capturing the dynamic interactions between patient conditions and treatments. To address this, we propose integrating reinforcement learning with a Generative Adversarial Network (GAN) and a dual-Hawkes process model to develop intelligent agents capable of delivering personalized and adaptive treatment strategies. The dual-Hawkes process allows us to model the intensity of both disease progression and recovery, while accounting for long-term dependencies. The GAN simulates real-world clinical environments using raw time-to-event data, without requiring detailed treatment annotations. By interacting with GAN, our model-based reinforcement learning agent learns an optimal dynamic policy that leverages long-term historical dependencies. When applied to the MIMIC-III dataset, our approach significantly increased the duration that patients remained in a healthy state, outperforming established machine learning policies.

## 1 INTRODUCTION

The patient's physical condition and illness can change rapidly during clinical treatment, primarily when doctors perform rescues in intensive care units (ICUs) (Goh et al., 2020). In such situations, providing accurate and timely targeted treatment to the patient can significantly alleviate the progression of the illness and may even save lives (Fleming & Harrington, 2013). Therefore, it is crucial to provide prompt and personalized treatment plans based on the patient's current physical state and the specific nature of their illness. Given reinforcement learning's excellent performance in various decision-making problems (Silver et al., 2016), training agents based on reinforcement learning (RL) to develop effective treatment strategies has been widely studied. However, recent works on RL-based dynamic treatment have some significant drawbacks.

Current methods rely solely on the final survival outcome to evaluate treatment plans and define a reward function accordingly (Coronato et al., 2020). This limitation prevents the agent from accurately responding to unexpected events during treatment and providing timely interventions (Yu et al., 2021). To address this issue, we propose a novel reward function based on the intensity function, which can promptly reflect the patient's disease onset tendency in real-time. The intensity function is the same as the hazard function in survival analysis (Fleming & Harrington, 2013), characterizes the likelihood of disease onset under specific conditions; it naturally indicates the patient's physical state. This enables our method to respond in real-time to changes in the patient's physical condition. In recent survival analysis research, the intensity is modeled as a function of current covariates (Fleming & Harrington, 2013). If we follow this paradigm, the reinforcement learning method with a reward based on intensity essentially relies on a Markov decision process (Natarajan & Kolobov, 2022). The assumption of a Markov decision process does not align with our real-world scenario. In actual medical settings, the future illness intensity should depend on the patient's history

of past occurrences and treatments. Therefore, one might consider building a higher-order Markov decision process (Zhou et al., 2023; Ye et al., 2024).

Considering the Hawkes process's demonstrated success in capturing long-term dependencies and modeling intensity functions (Meng et al., 2024b), we propose a *Dual-Hawkes* Process that combines the structures of both Hawkes (Lima, 2023) and Cox processes (Mei et al., 2024) to model the progression of disease, including both illness and recovery phases. This Dual-Hawkes model leverages the strengths of both processes, capturing the influence of historical events and current covariates on intensity simultaneously (Meng et al., 2024a).

To account for the recurring nature of disease events, we define two types of events: illness and recovery, and examine their respective intensity functions. The illness and recovery events are modeled using a multidimensional Hawkes process (Meng et al., 2024b), allowing us to simultaneously capture the effects of both historical events and covariates on the disease progression. Given the role of the intensity function in reflecting a patient's physical state, we define the difference between the recovery and illness event intensities over time as the reward function to train the agent. Since only recorded treatment data is available due to the high cost and scarcity of real medical data, the agent is trained using offline reinforcement learning. To enable the agent to interact with the environment and continuously propose treatment plans, we employ a Generative Adversarial Network (GAN) (Arjovsky et al., 2017) to iteratively generate synthetic clinical data (Kuo et al., 2022). In each iteration, the Dual-Hawkes process is used to compute the reward function, which is then used to train the agent effectively.

In summary, the main contribution of our work can be summarized as follows:

- We introduce a *Dual-Hwakes Process* model that integrates the Cox model and Hawkes process to capture the impact of a patient's physical condition and historical treatment on both the illness and recovering progressions, providing a realistic environment for agent interaction.
- We establish a model-based reinforcement learning framework embedded within a GAN and utilize the dual-Hwakes process so that agents can learn optimal treatment strategies with higher history dependencies.
- We apply the model to the MIMIC-III data, and the experiment result shows that our approach significantly increased the duration of patients remaining in a healthy state, outperforming current established machine learning policies.

## 2 DISEASE HAZARD MODELING WITH DUAL-HAWKES PROCESS

In many medical decision-making problems, the goal is to derive a dynamic treatment regime (DTR) (Chakraborty & Murphy, 2014) that provides a sequence of treatment decisions based on a patient's current health status and prior treatment history, aiming to improve long-term outcomes (Gottesman et al., 2019). Reinforcement learning (RL), due to its ability to optimize sequential decisions, has become increasingly popular in healthcare, particularly in DTR settings (Yu et al., 2021; Abdellatif et al., 2023; Coronato et al., 2020). For instance, several studies have applied RL to derive optimal treatment policies for conditions like sepsis, aiming to prevent critical events such as organ failure or death (Yu et al., 2021).

However, in many medical problems, the reward (outcome) is only observed or defined at the terminal state, for example, the patient survival (Komorowski et al., 2018; Tamboli et al., 2024). While this binary reward structure may seem straightforward, it oversimplifies the complexities inherent in clinical care. Such methods fail to account for important factors like early signs of deterioration, abnormal vital signs, and the progression of disease, all of which play critical roles in determining patient outcomes. Additionally, the sparse nature of terminal reward signals can exacerbate typical RL challenges, such as credit assignment and sampling inefficiency, leading to high variance in learning, as highlighted in Luo et al. (2024).

To address these limitations, some researchers have turned to continuous clinical risk scores, such as the Sequential Organ Failure Assessment (SOFA) score and the National Early Warning Score 2 (NEWS2) (Inada-Kim & Nsutebu, 2018). These scores have been integrated into RL models to provide more consistent and immediate rewards across time steps (Raghu et al., 2018; Wang et al.,

2022). However, these risk score-based methods still do not fully capture long-term dependencies and the cumulative effects of past clinical events, which are critical in effective healthcare decision-making. Hence, a more advanced approach would be to model the risk of adverse events, rather than relying solely on intermediate scores or terminal outcomes. With this motivation, we introduce a novel reward function based on a Dual-Hawkes Process, designed to model the intensities of both disease and recovering progressions over time.

The reward function is based on the difference between the intensities of recovering and disease illness events. By optimizing the reward function, the agent is trained to provide treatment plans to reduce the patient's disease illness intensity and increase the recovering intensity. Intensity function, also known as hazard function in survival analysis, is a function of current covariates in recent survival analysis work (Haarnoja et al., 2018). To capture long-term dependency from historical events, actions, and current covariates, we propose the Dual-Hawkes process, a combination of the Hawkes Process (Meng et al., 2024b) and the Cox model (Cox, 1972). Using our proposed Dual-Hawkes process to model the intensity function, the reward function can capture long-term dependencies by incorporating historical information (Wan et al., 2024).

## 2.1 DUAL-HAWKES PROCESS

In order to capture long dependency in recurring survival data and make informed decisions based on historical information, we have to model the intensity function based on current states and historical events. In recent studies, the Hawkes process (Meng et al., 2024b) has been used in modeling the intensity function based on historical events, and the Cox model (Cox, 1972) has been used in modeling the intensity function based on current covariates. To meet our requirements, we proposed a *Dual-Hawkes Processes* model to characterize disease occurrence and recovering processes. In this model, disease illness and recovering are conditioned on historical disease records, covariates representing the patient's physical condition, and historical treatment actions. In our proposed framework, the occurrence of diseases is depicted as a period of time on a continuous timeline. The beginning moment $\mathbf{tb}$ and the ending moment $\mathbf{te}$ of the disease are instant events on the timeline. The symptomatic time is the period between moment $\mathbf{tb}$ and $\mathbf{te}$. For simplicity of notation, assume that we observed a sequence of $m$ illness and recovering time points for a particular subject, denoted as $\{\mathbf{tb}_i, \mathbf{te}_i\}_{i=1}^m$. The intensity function, defined as the instantaneous event rate, is introduced to represent the illness and recovering mechanism of disease. Notably, the intensity function shares the same meaning as the hazard function in survival analysis (Fleming & Harrington, 2013). In this setting, we let $\lambda_1(t)$ and $\lambda_2(t)$ represent the instantaneous rate of event for the illness and recovering, respectively, at time $t$ conditioned on all historical information, including historical events, covariates, and treatment actions i.e.,

$$\lambda_k(t) = \lim_{t \to 0} \frac{\mathbb{E}[\mathrm{d}N_k(t)|\mathcal{H}_t]}{\mathrm{d}t}, \quad k \in \{1, 2\}, \ t \in [0, T], \tag{1}$$

where $\mathcal{H}_t$ denotes the historical information before time $t$, including historical illness, covariates and treatment actions, and $N_k(t)$ is the counting process of the corresponding event. The conditional intensity functions of the illness and recovering processes should then be modeled separately, while sharing partially the same historical information, such as the treatment history. Hence, we consider a new strategy called the Dual-Hawkes Process model that simultaneously model the two processes. Each of them is also inspired by the Hawkes Process for incorporating historical treatment events, and the Cox model for adjusting the hazard based on covariates. Formally, our conditional hazard function is defined as:

$$\lambda_k(t|\mathcal{H}_t) = (\mu_k + \sum_{t_i < t} \phi_{k,k_i}(t - t_i)) \exp(f_k(\mathbf{s}_t, \sum_{j=1}^{m_1} h(t - t'_{1_j}) \dots, \sum_{j=1}^{m_k} h(t - t'_{k_j}))), \tag{2}$$

where $\mu_k$ is the baseline intensity for event type $k$, $\mathbf{s}_t$ is the covariate information at time $t$, and $\phi_{k,k_i}(t - t_i)$ is the trigger kernel representing the excitation effect from event $t_i$ with type $k_i$ to $t$ with type $k$ and $k \in \{1, 2\}$. Moreover, $h(\cdot)$ is a Guassian kernel function with fixed parameter denoting the efficacy of the medication diminishes over time. Hence, consider $f_k(\cdot)$ as a function parameterized by neural networks for more flexibility, rather than the linear link used in the Cox model.

Traditional Hawkes Processes (Hawkes, 1971a) and Cox model (Cox, 1972) can both be considered as special cases of our proposed Dual-Hawkes Process. When we disregard the influence of covari-

ates and actions, set $\exp(f_k(\mathbf{s}_t, \sum_{j=1}^{m_1} h_{a_1}(t - t'_{1_j}) \ldots, \sum_{j=1}^{m_k} h_{a_k}(t - t'_{k_j})))$ as constant terms, the conditional intensity function can be written as:

$$\lambda_k(t|\mathcal{H}_t) = \mu_k + \sum_{t_i < t} \phi_{k,k_i}(t - t_i), \tag{3}$$

same as the traditional Hawkes Process's intensity function. Further, when we neglect the influence of historical events, setting $\mu_k + \sum_{t_i < t} \phi_{k,k_i}(t - t_i)$ as $\lambda(t)$, the baseline hazard, which is irrelevant to any historical event, the conditional intensity function can be written as:

$$\lambda_k(t|\mathcal{H}_t) = \lambda(t) \exp(f_k(\mathbf{s}_t, a_t)), \tag{4}$$

aligns with the hazard function for the Cox model if we treat the treatment at time $t$ as a part of covariates. Hence, our method can be viewed as a combination of Hawkes Process and Cox model.

With a series of observed illness and recovering events denoted as $\{\mathbf{tb}_i, \mathbf{te}_i\}_{i=1}^m$, all parameters in the intensity function are trained via maximum likelihood estimation. For a observed event sequence $\{\mathbf{tb}_i, \mathbf{te}_i\}_{i=1}^m$, the log-likelihood loss function is:

$$\mathbf{L} = \prod_{i=1}^m \lambda_1(\mathbf{tb}_i) \prod_{i=1}^m \lambda_2(\mathbf{te}_i) \exp\left(-\int_{T_1} \lambda_1(u)du\right) \exp\left(-\int_{T_2} \lambda_2(u)du\right). \tag{5}$$

In some real-world cases, disease illnesses are observed at discrete, fixed time points. The objective likelihood function is revised to the following form accordingly to adapt to these cases:

$$\mathbf{L} = \prod_{i=1}^m \mathbf{p}_1(\mathbf{tb}_i) \prod_{i=1}^m \mathbf{p}_2(\mathbf{te}_i) \exp\left(-\int_{T_1} \lambda_1(u)du\right) \exp\left(-\int_{T_2} \lambda_2(u)du\right), \tag{6}$$

where we have:

$$\mathbf{p}_1(\mathbf{tb}_i) = 1 - \exp\left(-\int_{\mathbf{te}_{i-1}}^{\mathbf{tb}_i} \lambda_1(u)du\right), \tag{7}$$

denoting the probability that the disease has not occurred until $\mathbf{tb}_i$ after $\mathbf{te}_{i-1}$ and $\mathbf{tb}_i$. At the same time,

$$\mathbf{p}_2(\mathbf{te}_i) = 1 - \exp\left(-\int_{\mathbf{tb}_i}^{\mathbf{te}_i} \lambda_2(u)du\right), \tag{8}$$

denoting the probability that the disease has not recovered until $\mathbf{te}_i$ after $\mathbf{td}_i$.

## 3 LEARNING DYNAMIC TREATMENT RULES

### 3.1 REINFORCEMENT LEARNING BASICS

We employ a **Reinforcement Learning (RL)** framework (Sutton, 2018) to train the agent responsible for generating treatment actions. In healthcare domains, RL is typically built upon the framework of **Markov Decision Process** (MDP, (Puterman, 1990)), denoted as $M = \{S, A, P, r, \gamma\}$, where $S$ is the state space, $A$ is the action space, $\gamma \in [0, 1)$ is the discount factor, $r : S \times A \to \mathbb{R}$ is the reward function, and $P : S \times A \to S$ represents the transition dynamics. The value function $v^\pi(s)$ is defined as the expectation of future discounted total rewards obtained by following a policy $\pi : S \to \Delta(A)$, expressed as: $v^\pi(s) = \mathbb{E}_\pi \left[\sum_{t=0}^\infty \gamma^t r(s_t, a_t) \mid s_0 = s\right]$, where $\mathbb{E}_\pi$ indicates the expectation under the policy $\pi$ and the transition probability. The corresponding action-value function is given by: $q^\pi(s, a) = r(s, a) + \gamma \mathbb{E}_{s' \sim P(\cdot|s,a)}[v^\pi(s')]$. The goal of RL is to find an optimal policy $\pi^*$ that maximizes the value for all $s \in S$.

Offline RL is proposed to train RL agents using pre-collected data instead of real-time interactions with the environment. This contrasts with online RL, where the agent learns through direct interaction with a real environment or a simulator (Zhou et al., 2024; Schulman et al., 2017). In an offline RL setting, the focus shifts to learning an optimal policy from a pre-gathered dataset. The dataset is assumed to result from actions taken according to a specific behavior policy $\pi_b$. A primary challenge

in offline RL is that $\pi_b$ may not adequately explore all possible actions, leading to potential over-optimistic estimation of out-of-distribution actions. Acting greedily concerning such actions could be problematic (Fujimoto et al., 2019). On the other hand, the Markov property of most existing RL models can be unrealistic in medical problems. To address this issue, we construct a modeling framework that would allow higher order dependencies on the historical state information (Zhou et al., 2023).

## 3.2 Embedding with Historical Information

In traditional MDP frameworks for DTRs, policies and reward structures tend to focus on the patient's immediate state, often resulting in decisions that overlook critical clinical history. To address this limitation, in our proposed framework, both the reward function and the transitional kernel depends on higher order history of the Markov process (Ye et al., 2024), while this dependency is adaptively learned in the dual-Hawkes process and the recurrent neural network (RNN). Instead of relying solely on the patient's current state, we employ a state representation embedded from an RNN, which encodes the patient's health trajectory over time. This richer embedding allows for decisions that incorporate a more comprehensive temporal context, considering both the current state and accumulated past information.

Specifically, within our framework, **Action $\mathbf{a}_t$** refers to the treatment vector, representing the dosages of various medications administered at time $t$. **Reward $\mathbf{r}_t$** is designed as the accumulated intensity function, formulated as:

$$\mathbf{r}_t = \int_t^{t+1} (\lambda_2(u) - \lambda_1(u)) du, \tag{9}$$

At time $t$, the history comprises the information $\mathcal{H}_{t-1} = \{\mathbf{s}_1, \ldots, \mathbf{s}_{t-1}, \mathbf{a}_1, \ldots, \mathbf{a}_{t-1}\}$. The **State** at time $t$, denoted as $\mathbf{s}_t$, is updated by the function:

$$\mathbf{s}_t = g(\mathcal{H}_{t-1}), \tag{10}$$

where $g(\cdot)$ is a recurrent neural network (RNN) that encodes the patient's prior health data and treatment history. Specifically, $\mathcal{H}_{t-1}$ is first transformed by an embedding layer into a vector $\mathbf{h}_t$, which is then used to update the state $\mathbf{s}_t$ by a linear layer. The treatment decision $\mathbf{a}_t$ is subsequently generated from $\mathbf{s}_t$ and $\mathbf{h}_t$ via a policy function $\pi(\cdot)$:

$$\mathbf{a}_t = \pi(\mathbf{s}_t, \mathbf{h}_t). \tag{11}$$

Both $g(\cdot)$ and $\pi(\cdot)$ are parameterized by neural networks. Furthermore, the RNN $g(\cdot)$ is trained within a Generative Adversarial Network (GAN) framework, which strengthens its ability to effectively capture and encode the patient's historical state information. Additional details on the GAN implementation are provided in the following section.

## 3.3 Training and Embedding Covariates with GAN

To ensure that our RNN-based model can effectively encode historical information, we employ a GAN framework to train $g(\cdot)$ when predicting the patient's state of future time. Specifically, the covariates $\mathbf{s}_t$ representing the patient's health state in period $t$ are generated by a bi-LSTM system (Zhang et al., 2015) based on historical information $\{\mathbf{s}_1, ..., \mathbf{s}_{t-1}\}$ and a sequence of specific treatment strategies $\{\mathbf{a}_1, ..., \mathbf{a}_{t-1}\}$, which can be either continuous or discrete variables representing the usage of multiple medications (Kuo et al., 2022). Given a random noise $\mathbf{z}$ drawn from a given distribution served as the initial state, a sequence of covariates $\hat{\mathbf{s}} = \phi(\mathbf{z})$ can be generated recursively by generator $\phi(\cdot)$ parasitized by a bi-LSTM. Further, given an observed real-world covariates sequence $\{\mathbf{s}_i\}_{i=1}^n$, the generator can be trained by the following minimax game between the generated data and the observed data:

$$\max_D \min_\phi \frac{1}{n} \sum_{i=1}^n (D(\phi(\mathbf{z}_i)) - D(\mathbf{s}_i)), \tag{12}$$

where $D(\cdot)$ is the discriminator function also parameterized by neural networks. The discriminator is trained to maximize the objective function while the generator is trained to minimize the objective

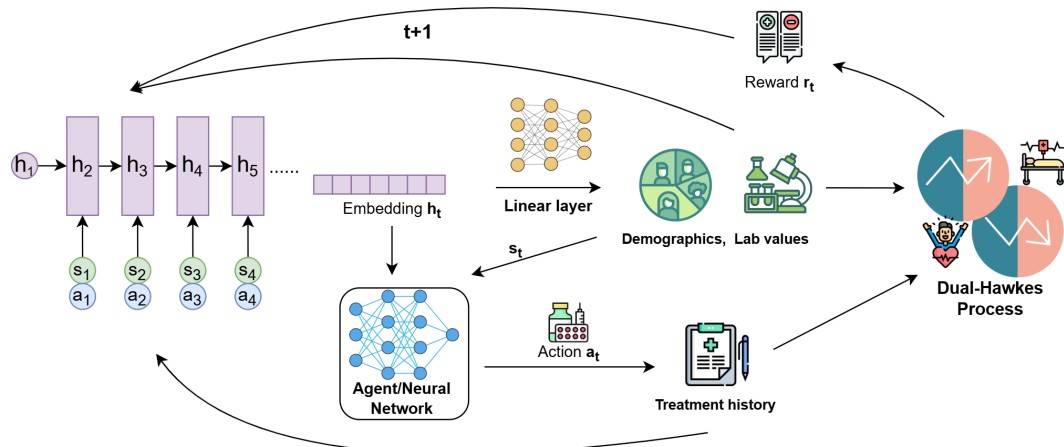

Figure 1: An overall framework of our proposed method: In this model-based reinforcement learning framework, the covariate sequence is generated by recurrent neural networks. The Dual-Hawkes process provides a reward function for training the agent, and the agent can make treatment decisions at every period.

function (Arjovsky et al., 2017). After proper adversarial training, the generator $\phi(\cdot)$ can generate covariates sequence recursively. Given generated historical covariates sequence $\{\mathbf{s}_1, ..., \mathbf{s}_{t-1}\}$ and specific treatment sequence $\{\mathbf{a}_1, ..., \mathbf{a}_t\}$, a well-trained generator can generate the next covariates states $\mathbf{s}_t$. This process simulates the progression of a patient's physical condition during treatment in a clinical setting.

The primary distinction of our approach within the RL framework lies in customizing the loss function. Instead of utilizing traditional loss functions, we employ our designed reward function to maximize the accumulated illness risk. This modification allows the agent to learn optimal treatment strategies tailored to effectively adapt to the evolving state of the patient's condition. During the training phase, the agent interacts with the simulated environment, continuously updating its policy to maximize the defined reward, thereby ensuring the provision of appropriate and personalized treatment plans across various cases. To train the agent, we simulate the patient's physiological environment and disease progression mechanisms, enabling the agent to learn through these interactions. The training objective is to optimize the decision functions $f(\cdot)$ and $g(\cdot)$ to maximize the cumulative reward. This optimization is achieved using gradient-based methods typically employed in training neural networks (Kingma, 2014), ensuring that the agent can effectively generalize its treatment strategies to diverse patient scenarios.

## 4 RELATED WORK

Temporal point processes can be used to model discrete event sequences over continuous time (Shchur et al., 2021). The Hawkes process is a widely used model for capturing event sequences (Hawkes, 1971b), where historical events influence subsequent occurrences (Hawkes, 1971a). With the development of deep learning, methods based on deep neural networks have been introduced for modeling event sequences (Meng et al., 2024b; Zuo et al., 2020). Furthermore, models that capture the impact of covariates on event occurrence have also been proposed, expanding the application scenarios of temporal point processes (Meng et al., 2024a). Interestingly, this scenario of considering the influence of covariates on event occurrence is quite similar to the Cox model in survival analysis (Fleming & Harrington, 2013) and can be viewed as two methods within a unified framework. In our proposed framework, the recurrence of diseases (Ren et al., 2019) is depicted by event sequences (Shchur et al., 2021) and modeled by a temporal point processes-based framework.

RL has emerged as a transformative approach in healthcare, particularly in the context of DTRs, where patient responses to treatments are used to adapt interventions over time (Coronato et al., 2020; Yu et al., 2021). Currently, methods mainly focus on Q-learning and value-based approaches (Komorowski et al., 2018; Luckett et al., 2020; Wu et al., 2023). And Luo et al. (2024) emphasizes

how results from RL algorithms, such as Conservative Q-Learning CQL, (Kumar et al., 2020) and Deep Q Networks DQN, (Mnih et al., 2015), can be inconsistent when different evaluation metrics or MDP formulations are applied. Furthermore, as Luo et al. (2024) argues, robust policy evaluation methods are lacking. Without reliable ways to assess how well the learned policies generalize to new patients, the effectiveness of these RL models remains uncertain. Moreover, current MDP-based methods are based on 1-order Markov assumptions (Bellman, 1957), contradicting the long-term dependency over time. Consequently, we must consider higher-order MDP (Ye et al., 2024), which is captured by our proposed Dual-Hawkes process in this framework.

## 5 EXPERIMENTS

### 5.1 SIMULATION STUDY

To evaluate the performance of the proposed *Dual-Hawkes Process* model, we conducted simulation experiments using generated data. Events representing illness (healthy-to-sick) and recovering (sick-to-healthy) were generated via the thinning algorithm, with intensity functions reflecting transition dynamics influenced by covariates. We designed three scenarios with different transition frequencies: weak, moderate, and strong transitions. Each scenario tests the model's ability to capture varying rates of state changes over time.

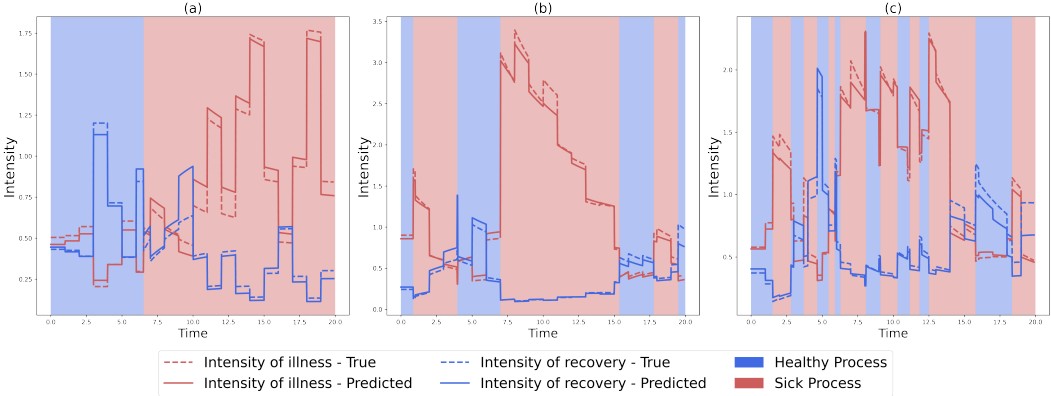

Figure 2: The experiment results in synthetic data demonstrate that in both healthy states and sick states, the Dual-Hawkes process can recover the ground truth intensity function.

For each scenario, we plotted the true intensity functions alongside the predicted intensity functions obtained from the Dual-Hawkes Process model, as shown in Figure 2. The results show that the Dual-Hawkes Process model effectively adjusts its intensity functions to match the frequency of state transitions across all settings. The accurate fitting of both scenarios demonstrates that our model is versatile and effective across varying frequencies of state changes.

### 5.2 DATA DESCRIPTION

We utilize the MIMIC-III database (Johnson et al., 2016), a comprehensive collection of de-identified clinical data from patients admitted to the Beth Israel Deaconess Medical Center in Boston. This publicly accessible database includes information from 53,423 adults and 7,870 neonate admissions, spanning over a decade, with a focus on critical care research. The dataset encompasses detailed demographic data, vital signs, laboratory test results, treatment protocols, and outcomes, along with free-text clinical notes.

Our analysis specifically targets the *sepsis* subset of the MIMIC-III database, which includes patients diagnosed with sepsis during their ICU stay. This dataset has been widely used for developing models aimed at sepsis detection and treatment, facilitating research into the management and outcomes of this critical condition. In this work, we use the IV fluid dosages and vasopressor dosages regime task for sepsis treatment, and categorize them into five classes, resulting in a $5 \times 5$ discrete

action space. The data pre-processing steps follow Komorowski et al. (2018). For the standardization of variables, we partitioned numeric variables with significantly long-tailed distributions into deciles and treated them as categorical variables. For the remaining continuous numeric variables, we applied logarithmic transformation when appropriate, followed by centering and scaling within the range [0,1] using the *MinMaxScaler*, which are consistent with Kuo et al. (2022). For the definition of the patients' healthy state and sick state, which indicates the transition events of recovery and illness, we refer to the SOFA score (Kajdacsy-Balla Amaral et al., 2005; Jones et al., 2009), a commonly used critical care metric to quantify the severity of a patient's organ function or rate of failure. And we define a patient as being in a sick state at each time point if he/she fulfills at least one of the following two conditions at that point:

- The SOFA score at that time point is higher than 4 (average of all patient records) and the SOFA score does not decrease more than 3 (average of all decreasing records) from the previous time point.

- The SOFA score at that time point is not higher than 4 (average of all patient records), but the rise in SOFA score from the previous time point is greater than or equal to 3 (average of all rising records)

For the definition of rewards in RL for sepsis treatment, the initial reward design proposed by Komorowski et al. (2018) utilizes a binary reward scheme: $r = 0$ for non-terminal steps, $r = +100$ for patient survival, and $r = -100$ for death at the final step. While straightforward, this approach oversimplifies the complexity of medical treatment scenarios, ignoring critical factors like the patient's risk of deterioration, abnormalities in vital signs, and disease progression rates, all of which significantly impact mortality. From a reinforcement learning perspective, such sparse and binary rewards can exacerbate challenges related to credit assignment, sampling inefficiency, and high learning variance.

In contrast, incorporating intermediate rewards has proven effective in goal-reaching RL tasks (Zhai et al., 2022). In the context of DTRs, intermediate rewards often take the form of clinical risk scores. For example, the SOFA score has been widely used in reward design for medical RL applications (Raghu et al., 2018; Wang et al., 2022). Additionally, lactate levels, which serve as biomarkers for tissue hypoxia and metabolic dysfunction (Nguyen et al., 2004), have been integrated into reward structures.

However, these risk-based rewards fail to fully account for the effects of historical events and the long-term dependencies critical in medical decision-making. Our proposed reward design, based on the intensity of the dual Hawkes process, addresses these limitations by capturing both immediate and historical factors, offering a more nuanced and dynamic reward framework.

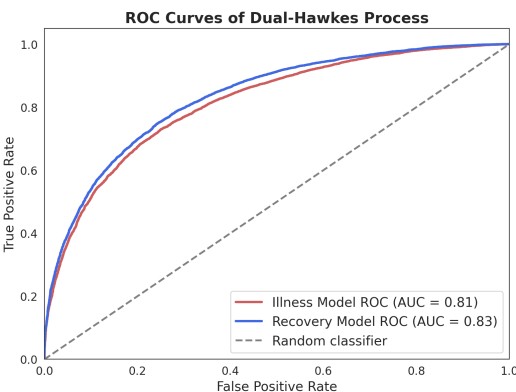

Figure 3: ROC Curves of Dual Hawkes Models

## 5.3 FITTING DUAL-HAWKES PROCESS

As previously discussed, our dataset comprises discrete observations, including covariates, actions (dosages of intravenous fluids and vasopressors), and transition events related to recovering and illness for each patient at various time points. To account for the time-varying nature of the covariates, we utilized the most recent time point's covariates to approximate their values at any given time $t$. This pragmatic approach allows us to capture the progression of clinical events more effectively. By integrating these covariates into an adapted likelihood function for discrete data, we can predict the probability of state transitions within the subsequent time slot based on the current data, thus leveraging the sequential dynamics inherent in patient care.

| Model | Accuracy | F1-score |
|---|---|---|
| Illness Model | 74.38% | 69.19% |
| Recovering Model | 75.23% | 76.16% |

Table 1: Model Performance Metrics

To further evaluate the performance of our fitted Dual-Hawkes Process model, we applied it to a test dataset to forecast the likelihood of transition events at each time point. The results were visualized through ROC curves (Figure 3), where the illness model achieved an AUC of 0.81, and the recovering model outperformed slightly with an AUC of 0.83, demonstrating the robustness of the intensity models in predicting state changes. Additionally, we compared the model's performance through standard classification metrics, as shown in Table 1. The illness model yielded an accuracy of 74.38% and an F1-score of 69.19%, while the recovering model showed superior results with an accuracy of 75.23% and an F1-score of 76.16%.

## 5.4 POLICY EVALUATION

This section outlines our selection of baseline models and policy evaluation methods. In reviewing the existing literature, particularly the work of Luo et al. (2024), we identified several gaps in the current methodology: varying baseline sets across studies, a lack of state-of-the-art (SOTA) offline RL algorithms, and the absence of naive baselines for essential evaluations. This reference served as a guiding framework for our baseline selection, emphasizing the importance of rigorous comparisons in assessing the effectiveness of RL approaches.

To address these identified issues, we incorporated a comprehensive set of baselines. We employed naive baselines, including zero-drug, random, and max-drug policies, to provide a foundation for comparison. In particular, GAN models, which were used as supervised learning baselines, feature a generator that leverages a Long Short-Term Memory (LSTM) network. This allows the GAN to capture behavior policies from the training data and assess the performance of RL against learned behavioral strategies, further enriching our analysis.

Furthermore, we included advanced RL baselines, specifically CQL (Conservative Q-Learning) and DQN (Deep Q-Network). These models were trained using clinical risk scores, as discussed earlier, with the SOFA score being a prominent example. However, as noted in Luo et al. (2024), existing RL methods like CQL and DQN may exhibit performance inferior to even naive baselines, such as the max-drug policy, when rewards are altered. This phenomenon was also observed in our experimental results, where both CQL and DQN underperformed in certain cases when the reward function was modified. By comparing our results against this diverse set of baselines, we aim to demonstrate the advantages of our approach in optimizing dynamic treatment regimes, particularly in the context of sepsis management.

Policy evaluation in RL for DTRs presents significant challenges due to the nature of the data (Luo et al., 2024). Since the dataset is fixed and observational, RL models cannot be evaluated through direct interaction with the environment, as is common in traditional RL settings. This limitation is compounded by the complexity of medical decision-making, where treatment effects may not be immediately apparent and are subject to a range of confounding variables. The variability in patient responses to treatments further complicates efforts to assess the effectiveness of proposed policies.

Several methods have been developed to address these challenges in offline RL, including Inverse Probability Weighting (IPW) (Liu et al., 2017), Weighted Importance Sampling (WIS) (Kidambi et al., 2020; Nambiar et al., 2023), the Direct Method (DM) (Huang et al., 2022; Kondrup et al., 2023), and Doubly Robust (DR) (Raghu et al., 2017; Wu et al., 2023; Wang et al., 2018), estimators. These approaches attempt to tackle the issue of confounding variables by creating counterfactual estimations based on historical data. However, no single method has proven universally effective, and each has its own limitations, particularly in accounting for the complexities of medical treatment data.

Our framework addresses these limitations by providing a virtual environment to simulate policy interactions with patients, thus offering a more dynamic evaluation platform. Specifically, we leverage a combination of GAN and Dual-Hawkes Process models to simulate patient responses under different treatment policies. By simulating the average intensity of recovering and illness progression (i.e., the integral of recovering intensity minus illness intensity), we can compare the performance of different policies in a more nuanced way. Additionally, the simulation estimates the total time patients spend in a healthy state, also offering a measure of policy effectiveness (shown in Figure 5

in Appendix). During the agent training process, we use the difference between the recovering and illness intensity integrals as the reward to actions.

## 5.5 RESULTS

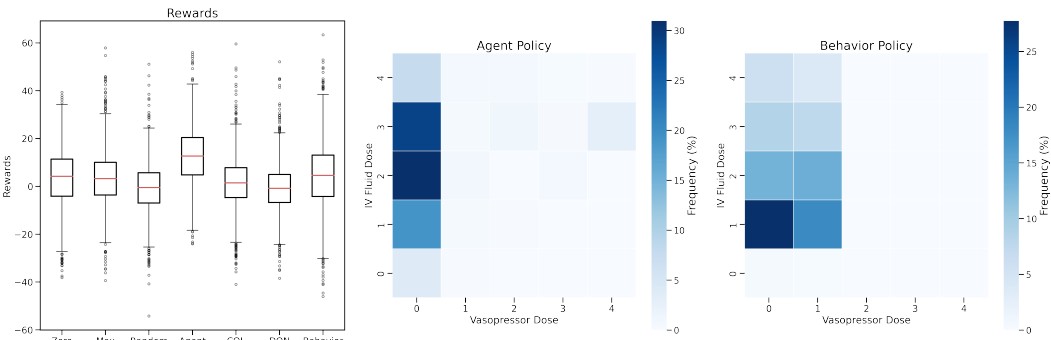

Figure 4: Comparison of policies' performance and action distributions.

We followed the outlined approach to train our reinforcement learning agent and evaluated its performance within the comparative framework described earlier. Specifically, we compared the effectiveness of all methods in terms of the difference in the integral values of recovering intensity and illness intensity, as shown in Figure 4. Our method outperformed all baseline approaches, showing a higher average performance and lower variance.

Among the tested approaches, the agent's performance was closest to the behavior policy, indicating that the agent successfully learned from the historical data while improving on it. Notably, DQN and CQL underperformed compared to simpler policies like the Max-drug and Zero-drug strategies, a finding consistent with observations from Luo et al. (2024). This highlights the sensitivity of existing RL methods to reward structures, which can lead to worse performance compared to naive baselines.

In addition, we analyzed the distribution of actions taken by the agent compared to the behavior policy. The primary distinction we observed was that our policy increased the dosage of IV fluids while reducing the usage of vasopressors, compared to the behavior policy. This adjustment resulted in a significant improvement in patient outcomes, as evidenced by an increased reward in terms of prolonged healthy state durations and improved health status overall.

These results suggest that our agent effectively optimized the treatment strategy, leading to improved patient outcomes, and more favorably balanced recovering and illness intensities, while maintaining a stable action distribution that aligns with real-world clinical decisions. By reducing vasopressor usage and moderately increasing IV fluids, our method notably enhanced patient recovering, improving the reward over the behavior policy's supervised learning strategy.

## 6 CONCLUSION

In this study, we introduced the *Dual-Hawkes Process*, which integrates Cox models and Hawkes processes to simultaneously model disease illness and recovering events while accounting for historical events and covariates. Coupled with a novel model-based reinforcement learning framework that employs a GAN for offline training, our approach utilizes the difference in intensity functions as a reward signal to train agents, which provides timely and preventive treatment. Applied to the MIMIC-III dataset, our method significantly extended the duration for which patients remained healthy compared to existing reinforcement learning policies. This work overcomes the limitations of previous approaches by providing more accurate real-time state modeling and multi-outcome optimization, paving the way for more effective and personalized treatment strategies in critical healthcare settings.

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

# A APPENDIX

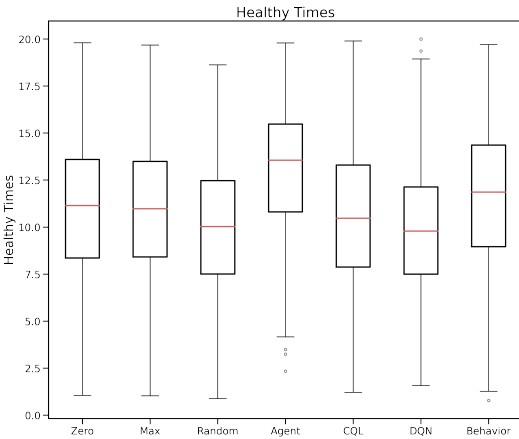

Figure 5: Healthy Time

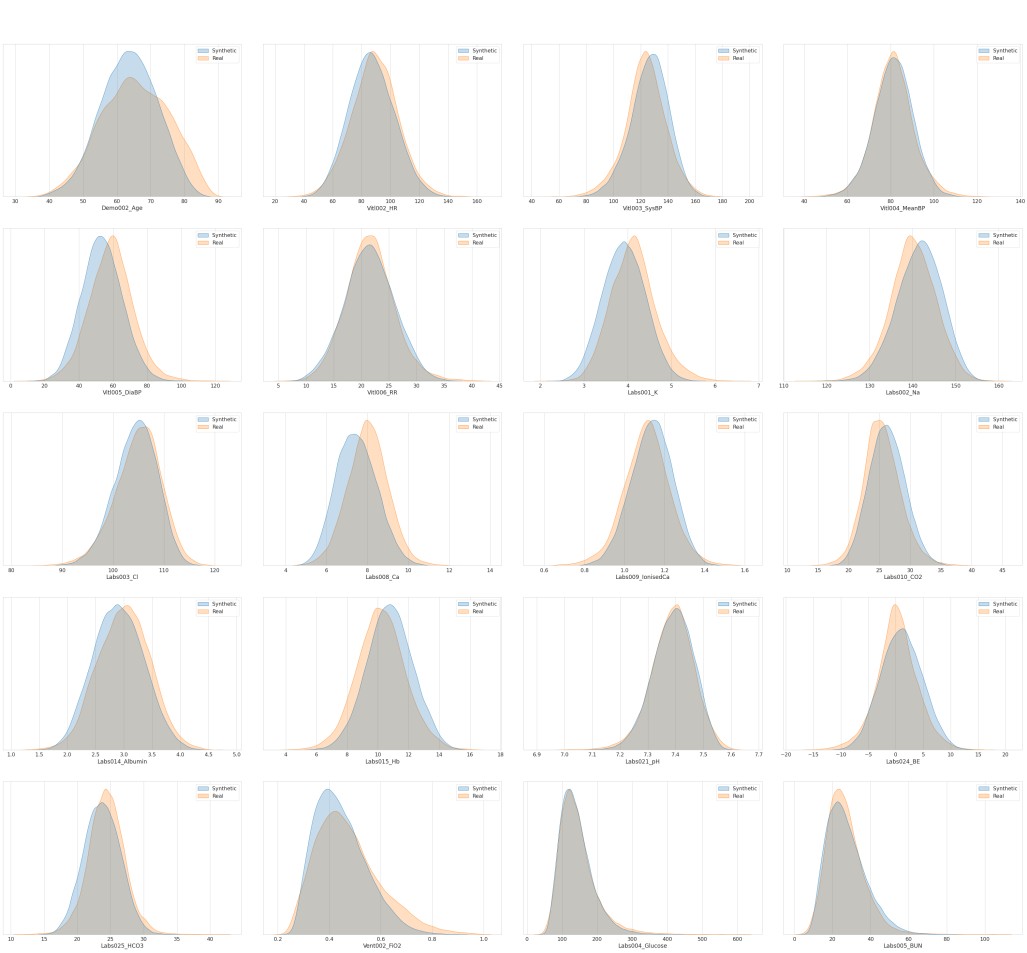

Figure 6: Distribution Plots for Sepsis 01

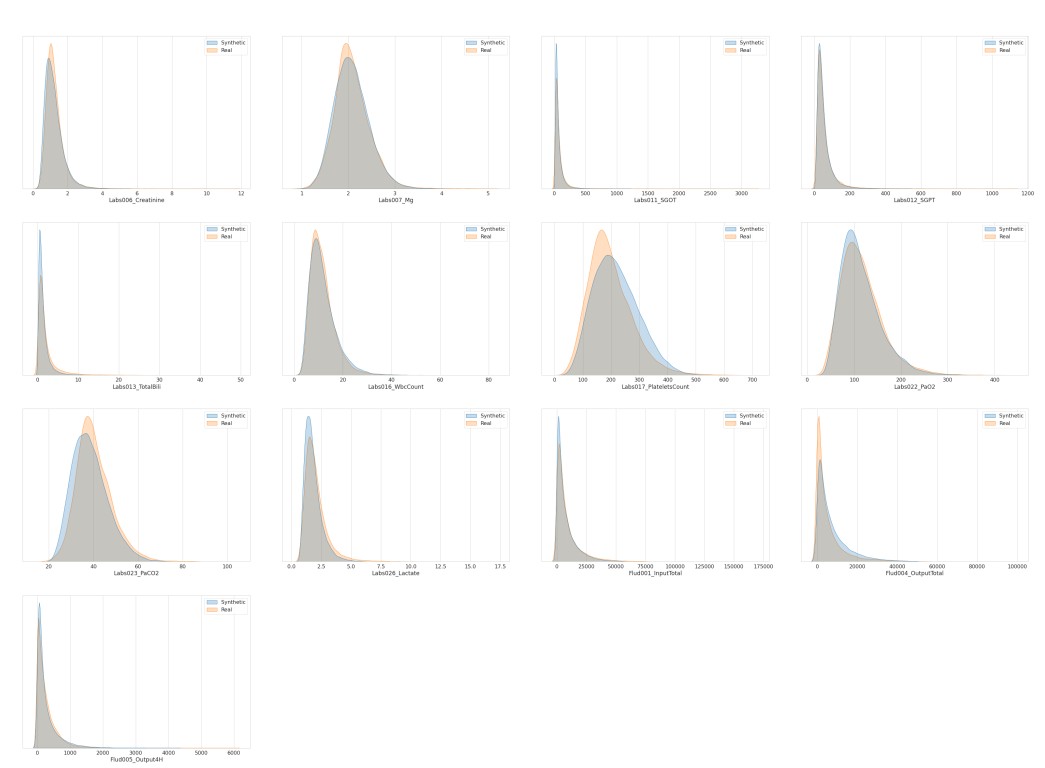

Figure 7: Distribution Plots for Sepsis 02

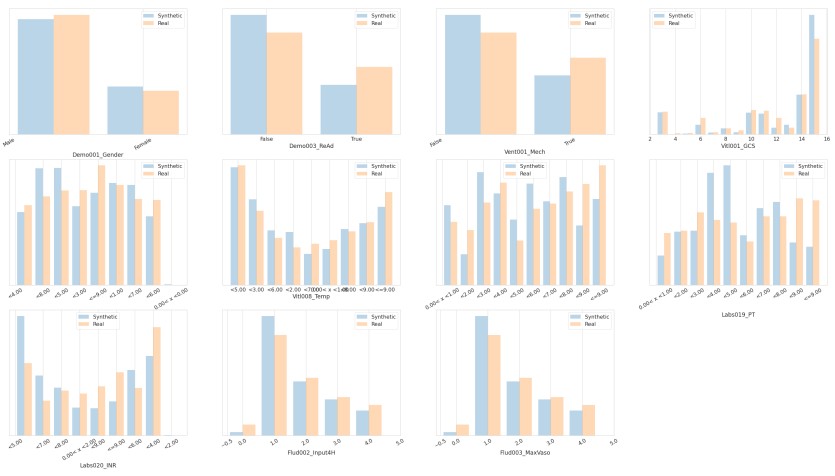

Figure 8: Distribution Plots for Sepsis 02

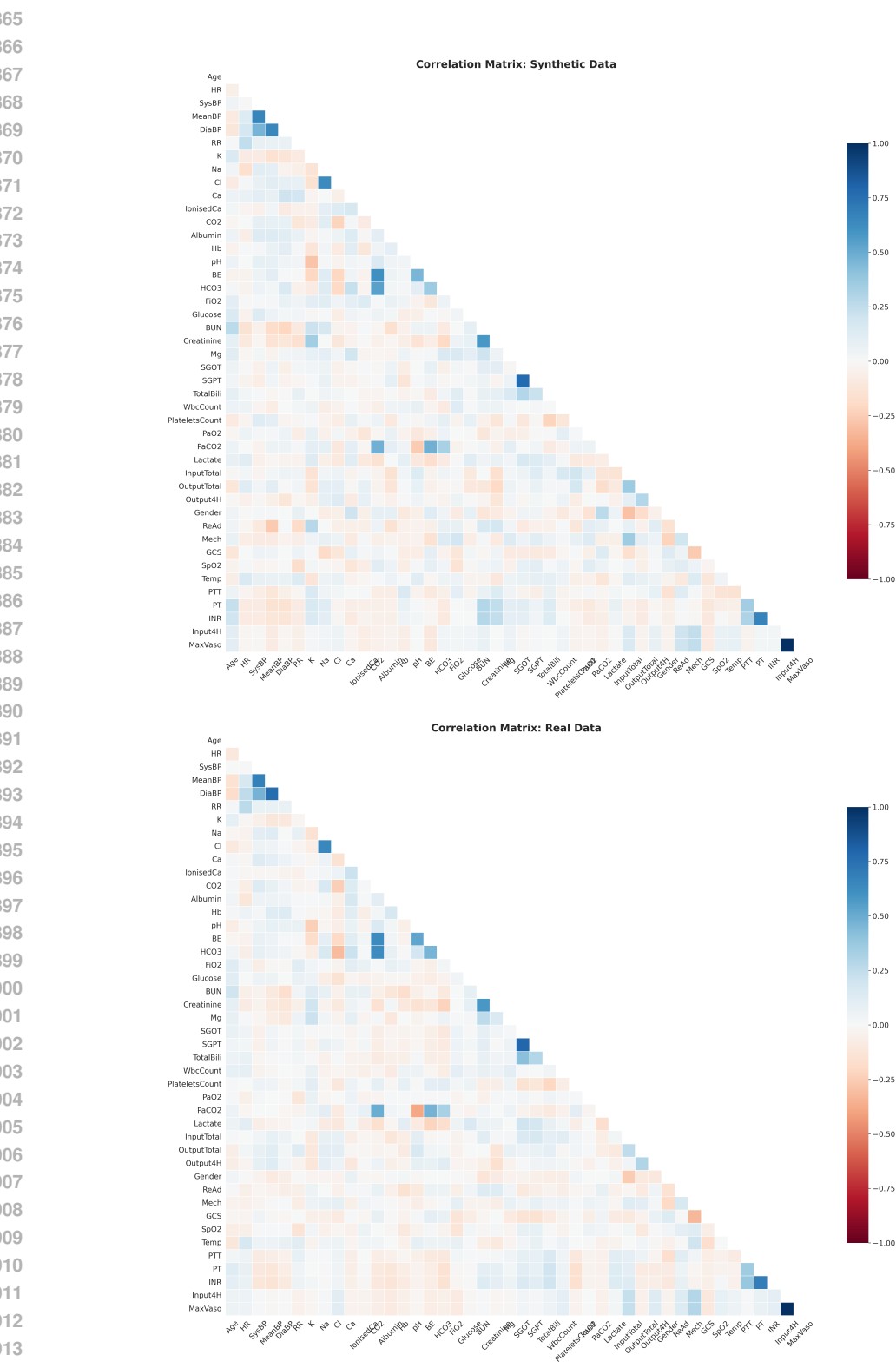

Figure 9: The Complete Static Correlation Plots for Sepsis

