# OpenReview forum: "Optimizing Dynamic Treatment Strategies with Reinforcement Learning and Dual-Hawkes Process in Clinical Environments"
_ICLR.cc/2025/Conference — Submitted to ICLR 2025_

### Official Review · Reviewer_H1oF · 2024-10-31

**Soundness:** 1
**Presentation:** 1
**Contribution:** 1
**Rating:** 3
**Confidence:** 3

**Summary:**

The paper introduces a Dual-Hawkes approach to model the reward function for offline reinforcement learning of dynamic treatment strategies using time-series event data. The Dual-Hawkes approach combines the Hawkes process and the Cox process to characterize the intensity functions for recovery and illness, wherein the reward function is specified as the difference between these functions. Experimental results on synthetic and semi-synthetic datasets demonstrate that the Dual-Hawkes model accurately recovers the simulated intensity function.

**Strengths:**

- The paper addresses an important problem of learning a dynamic treatment strategy from time-series event data.
- The proposed Dual-Hawkes approach seems to recover the simulated intensity function effectively.

**Weaknesses:**

- Given that the novelty seems quite limited to the proposed Dual-Hawkes process for modeling time-series event data, the paper does not discuss the proposed approach in the context of several competitive related works, including [1, 2, 3]. Without benchmarks against these competitive approaches, it is difficult to assess the significance of the proposed approach.

- The proposed framework appears quite complex with a combination of various approaches, with little motivation or justification for the modeling choices. For example, the decision to use an RNN versus a Transformer for modeling $g(\cdot)$, training with or without the GAN objective, and the choice of the Gaussian kernel in Equation 2 are not clearly justified.

- The paper must be improved for clarity. Several sections are difficult to follow; for instance, the functions $f(\cdot)$, $\pi(\cdot)$, and $\phi(\cdot)$ are left unspecified. It is unclear how the reward function using the MIMIC-II dataset is simulated and how the model is trained end-to-end. The paper introduces various aspects of the model and does not tie them together to create a comprehensive view, making it difficult to follow.

- The proposed RL approach for treatment recommendation does not account for time-varying confounding, which could bias the treatment assignments given observational data.


**References**
-  [1] Yamac et al. (2023),  "Hawkes Process with Flexible Triggering Kernels", MLHC
-  [2] Simiao et al. (2020), "Transformer hawkes process", ICML
-  [3] Oleksandr et al. (2020),  " Intensity-free learning of temporal point processes", ICLR

**Minor**
- Typos: Hwakes -> Hawkes, dual-Hawkes -> Dual-Hawkes
- Notation overload: $\phi(\cdot)$  used for both the intensity triggering kernel and the GAN generator

**Questions:**

- Figure 4: What are the agent and behavioral approaches? How is the ground truth reward function simulated?
- How are the functions $f(\cdot)$, $\pi(\cdot)$, and $\phi(\cdot)$ parameterized?
- How is the model trained end-to-end?
- Could you benchmark the Dual-Hawkes approach against competitive baselines for modeling time-series event data?
- Could you provide ablation studies or motivations justifying the various modeling choices?

---

### Official Review · Reviewer_QicS · 2024-11-04

**Soundness:** 2
**Presentation:** 3
**Contribution:** 2
**Rating:** 3
**Confidence:** 5

**Summary:**

This paper proposes a model-based reinforcement learning method that incorporates ideas from survival analysis to define the rewards, specifically Hawkes process and Cox model.

**Strengths:**

- Interesting combination of ideas from RL and survival analysis.
- Experiments conducted on both a simulated environment and a real dataset (MIMIC-III).

**Weaknesses:**

- The main contribution of the paper is unclear. In Sec 3, there is a part about policy learning that relies on a modified reward function based on the Dual-Hawkes Process model, and there is another part on training a GAN to learn history embeddings. Both are presented as the proposed method for learning policies, yet in the MIMIC-III experiment the GAN part is only used as a simulator in the off-policy evaluation. Which one is the main contribution?
- There are several inconsistencies or lack of clarity in the methods and experiments description (see questions below).

**Questions:**

- Why model disease and recovery process separately? In the data, there’s only a sparse reward signal. How are these two processes decoupled?
- The method section does not describe the order of training. Do you first learn the disease/recovery processing then use them as rewards (L66 - by taking the difference - this was actually never mentioned again in the paper), or do you train jointly?
- Figure 1 is never reference or explained. When should the reader look at the figure relative to the text sections?
- L297 why do we want to "**maximize** the accumulated illness risk"?
- L388 seems to suggest that the trajectory data needs to be first segmented into disease process vs recovery process, but this is not described anywhere in the methods.
- L388 the disease/recovery threshold for SOFA score seems arbitrary without proper reference/justification. Also there are no sensitivity analysis that explores different disease/recovery thresholds and their impact on the results.
- Fig 5 what are the error bars showing?
- Sec 5.5 there are several unsubstantiated medical claims without any supporting references. "By reducing vasopressor usage and moderately increasing IV fluids, our method notably enhanced patient recovering" - is this statement medically correct and how does your data support this statement? What types of patients do these changes apply to?

---

### Official Review · Reviewer_FbHn · 2024-11-05

**Soundness:** 2
**Presentation:** 2
**Contribution:** 2
**Rating:** 3
**Confidence:** 4

**Summary:**

This paper proposes dual-Hawkes process to model the reward function and train a policy network that learns optimal treatment strategies. The dual-Hawkes process is learnt from offline data, the transition dynamics are modeled also using offline data to create a model-based environment for RL training.

**Strengths:**

To the best of my knowledge, using dual-Hawkes process to model disease progression and recovery and using it as RL reward have not been proposed in prior works, so this work has some novelty.

**Weaknesses:**

1. I have doubt regarding the usage of model-based + policy-optimization-based RL with only offline data, and I do not think the authors have adequately provided analytical & experimental evidence to justify this design. Proper model-based RL with policy optimization still entails online update of the transition dynamics model, and if you do not perform such update, the drawback of offline RL you described in L215-217 and L472-479 still exist in the proposed method, because the "virtual environment to simulate policy interactions" (L480-481) you established is in-distribution to offline data. Analytically, CQL and importance-weight-based methods were specifically designed to tackle this problem, but I did not find any motivation or valid argument to support shifting away from these method to use a model-based + policy-optimization-based design without further adjustment. Experimentally, I did not find any valid evidence that shows DQN (including CQL) underperforms compared to such design either. See the next paragraph for more details.

2. In my judgement, this work has a very weak experiments section for a non-theoretical paper. Most content in this section are dedicated to transition dynamics model evaluation (Sec 5.1-5.3) and literature review on offline RL (Sec 5.4), the actual results on policy evaluation are barely more than half of a page (L489-525). I understand dual-Hawkes process is your main contribution, but the context of the paper is the usage of dual-Hawkes process as RL reward and it needs to be evaluated in the relevant context -- more energy should be put into assessing how using such reward modeling enhances the performance of policy optimization rather than how accurate it is on offline data. More importantly, I do not think the results on policy evaluation are fair to CQL and DQN, since DQN and CQL are trained on clinical risk scores (that's what I'm getting from L458-463, if this is not the case, please clarify), while the proposed RL framework is trained on the integral values of modeled recovering intensity and illness intensity, and all methods are evaluated on the integral values of modeled recovering intensity and illness intensity. Since DQN and CQL are optimizing a different target, this does not:
 - prove that the proposed model-based + policy-optimization-based RL design is better than CQN and DQN on offline data
 - prove that "the sensitivity of existing RL methods" (L514) does not exist in the proposed method

and most importantly, it also does not prove that using a pretrained dual-Hawkes process enhance the performance of offline policy optimization because you're treating your simulator as the truth by doing this, and using this simulator for both training and evaluation essentially makes your proposed method an online learning algorithm, so you're basically comparing your online method to other offline methods.

I think the following experimental designs would be a lot more convincing:
 - Evaluate the proposed method on a simulation dataset where a simulator is accessible only for generating offline training data and evaluating trained model, while not directly interacting with model training.
 - Conduct ablation studies comparing model-based + policy-optimization-based RL with CQN and DQN, with all of them using the same reward function.
 - Conduct ablation studies comparing dual-Hawkes process reward and clinical risk score reward on the same RL model, and the evaluation metric should not be the exact same as either reward models used in training.

**Questions:**

1. L451-457: I'm confused by L454 shifting to the description of GAN. How is it related to description of baselines?

2. L204, L225-226: Include some citations to back up the claim that prior work tend to assume Markov? I doubt if this is true without citations because Q-learning can be extended to non-Markovian formulation fairly easily.

3. L326-328: Relevance? Does this imply that you claim the proposed method entails a more robust policy evaluation method?

4. I think more detailed descriptions (equations, loss functions, etc.) of the RL training procedure should be presented. If pages are not enough, you could reduce Sec 5.4 to do so. Sec 5.4 is pretty out-of-place in the experiment section anyway.

---

### Meta-Review · Area_Chair_2cq7 · 2024-12-21

**Metareview:**

This paper proposes a reward model using the dual-Hawkes process to train a model based RL for disease progression modeling in healthcare data. Model-based approach uses GANs as a simulator. Evaluation on MIMIC data demonstrates reasonable performance. Reviewers have raised multiple concerns i) about the choice of GAN as simulator, ii) lack of clarity on core contributions, and iii) insufficent empirical evaluation. No author response was provided. Based on reviewer consensus, I recommend a reject.

**Additional Comments On Reviewer Discussion:**

Authors did not respond to reviews with a rebuttal. Therefore there was no discussion and changes during the rebuttal period.

---

### Decision · Program_Chairs · 2025-01-22

Reject